# LOGAN: LATENT OPTIMISATION FOR GENERATIVE ADVERSARIAL NETWORKS

## ABSTRACT

Training generative adversarial networks requires balancing of delicate adversarial dynamics. Even with careful tuning, training may diverge or end up in a bad equilibrium with dropped modes. In this work, we introduce a new form of latent optimisation inspired by the CS-GAN and show that it improves adversarial dynamics by enhancing interactions between the discriminator and the generator. We develop supporting theoretical analysis from the perspectives of differentiable games and stochastic approximation. Our experiments demonstrate that latent optimisation can significantly improve GAN training, obtaining state-of-the-art performance for the ImageNet ($128 \times 128$) dataset. Our model achieves an Inception Score (IS) of $148$ and an Fréchet Inception Distance (FID) of $3.4$, an improvement of $17\%$ and $32\%$ in IS and FID respectively, compared with the baseline BigGAN-deep model with the same architecture and number of parameters.

## 1 INTRODUCTION

Generative Adversarial Nets (GANs) are implicit generative models that can be trained to match a given data distribution. GANs were originally proposed and demonstrated for images by Goodfellow et al. (2014). As the field of generative modelling has advanced, GANs have remained at the frontier, generating high-fidelity images at large scale (Brock et al., 2018). However, despite growing insights into the dynamics of GAN training, most recent advances in large-scale image generation come from architectural improvements (Radford et al., 2015; Zhang et al., 2019), or regularisation focusing on particular parts of the model (Miyato et al., 2018; Miyato & Koyama, 2018). Inspired by the compressed sensing GAN (CS-GAN; Wu et al., 2019), we further exploit the benefit of latent optimisation in adversarial games using natural gradient descent to optimise the latent variable $z$ at each step of training, presenting a scalable and easy to implement approach to improve the dynamical interaction between the discriminator and the generator. For clarity, we unify these approaches as latent optimised GANs (LOGAN).

To summarise our contributions:

1. We present a novel analysis of latent optimisation in GANs from the perspective of differentiable games and stochastic approximation (Balduzzi et al., 2018; Heusel et al., 2017), arguing that latent optimisation can improve the dynamics of adversarial training.

2. Motivated by this analysis, we improve latent optimisation by taking advantage of efficient second-order updates.

3. Our algorithm improves the state-of-the-art BigGAN-deep model (Brock et al., 2018) by a significant margin, without introducing any architectural change or additional parameters, resulting in higher quality images and more diverse samples (Figure 1 and 2).

## 2 BACKGROUND

### 2.1 NOTATION

We use $\theta_D$ and $\theta_G$ to denote the vectors representing parameters of the generator and discriminator. We use $x$ for images, and $z$ for the latent source generating an image. The prime $'$ is used to denote

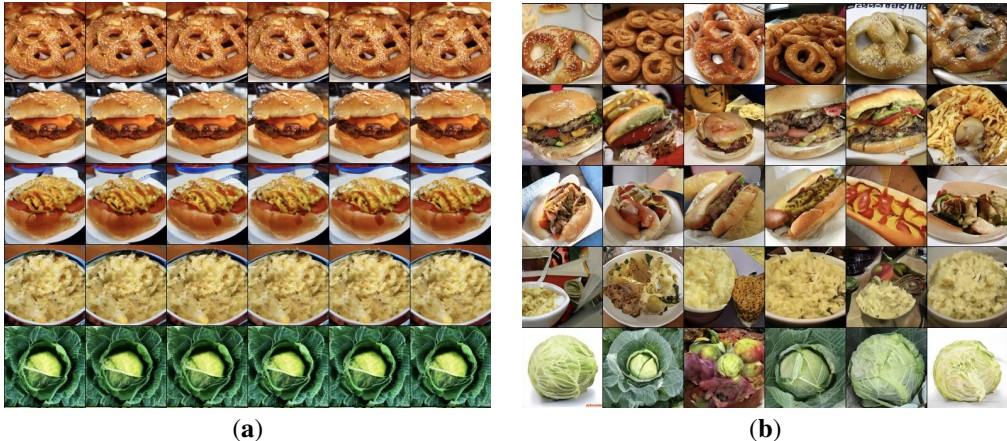

Figure 1: Samples from BigGAN-deep (**a**) and LOGAN (**b**) with similarly high IS. Samples from the two panels were drawn from truncation levels corresponding to points C and D in figure 3 **b** respectively. (FID/IS: (**a**) 27.97/259.4, (**b**) 8.19/259.9)

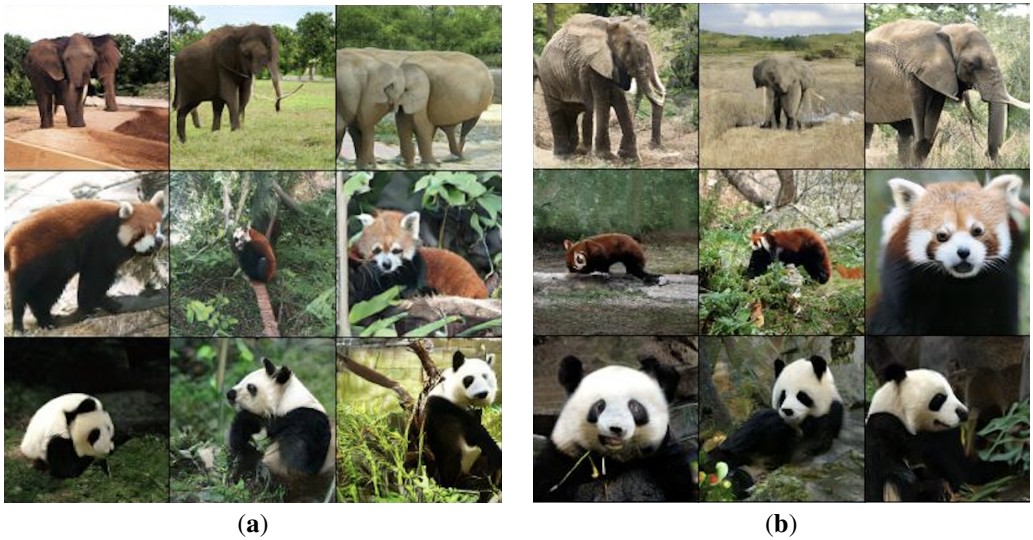

Figure 2: Samples from BigGAN-deep (**a**) and LOGAN (**b**) with similarly low FID. Samples from the two panels were drawn from truncation levels corresponding to points A and B in figure 3 **b** respectively. (FID/IS: (**a**) 5.04/126.8, (**b**) 5.09/217.0)

a variable after one update step, e.g., $\theta'_D = \theta_D - \alpha \frac{\partial f(z; \theta_D, \theta_G)}{\partial \theta_D}$. $p(x)$ and $p(z)$ denote the data distribution and source distribution respectively. $\mathbb{E}_{p(x)}\left[f(x)\right]$ indicates taking the expectation of function $f(x)$ over the distribution $p(x)$.

## 2.2 GENERATIVE ADVERSARIAL NETS

A GAN consists of a generator that generates image $x = G(z; \theta_G)$ from a latent source $z \sim p(z)$, and a discriminator that scores the generated images as $D(x; \theta_D)$ (Goodfellow et al., 2014). Training GANs involves an adversarial game: while the discriminator tries to distinguish generated samples $x = G(z; \theta_G)$ from data $x \sim p(x)$, the generator tries to fool the discriminator. This procedure can be summarised as the following min-max game:

$$\min_{\theta_D} \max_{\theta_G} \mathbb{E}_{z \sim p(x)}\left[h_D(D(x; \theta_D))\right] + \mathbb{E}_{z \sim p(z)}\left[h_G(D(G(z; \theta_G); \theta_D))\right] \tag{1}$$

Table 1: Comparison of model scores. BigGAN-deep results are reproduced from Brock et al. (2018). "baseline" indicates our reproduced BigGAN-deep with small modifications. The 3rd and 4th columns are from the gradient descent (GD, ablated) and natural gradient descent (NGD) versions of LOGAN respectively. We report the Inception Score (IS, higher is better, Salimans et al. 2016) and Fréchet Inception Distance (FID, lower is better, Heusel et al. 2017).

|     | BigGAN-Deep | baseline | LOGAN (GD) | LOGAN (NGD) |
| --- | --- | --- | --- | --- |
| FID | $5.7 \pm 0.3$ | $4.92 \pm 0.05$ | $4.86 \pm 0.09$ | $\mathbf{3.36 \pm 0.14}$ |
| IS | $124.5 \pm 2.0$ | $126.6 \pm 1.3$ | $127.7 \pm 3.5$ | $\mathbf{148.2 \pm 3.1}$ |

The exact form of $h(\cdot)$ depends on the choice of loss function (Goodfellow et al., 2014; Arjovsky et al., 2017; Nowozin et al., 2016). To simplify our presentation and analysis, we use the Wasserstein loss (Arjovsky et al., 2017), so that $h_D(t) = -t$ and $h_G(t) = t$. Our experiments with BigGAN-deep uses the hinge loss (Lim & Ye, 2017; Tran et al., 2017), which is identical to this form in its linear regime. Our analysis can be generalised to other losses as in previous theoretical work (e.g., Arora et al. 2017). To simplify notation, we abbreviate $f(z; \theta_D, \theta_G) = D(G(z; \theta_G); \theta_D)$, which may be further simplified as $f(z)$ when the explicit dependency on $\theta_D$ and $\theta_G$ can be omitted.

Training GANs requires carefully balancing updates to $D$ and $G$, and is sensitive to both architecture and algorithm choices (Salimans et al., 2016; Radford et al., 2015). A recent milestone is BigGAN (and BigGAN-deep, Brock et al. 2018), which pushed the boundary of high fidelity image generation by scaling up GANs to an unprecedented level. BigGANs use an architecture based on residual blocks (He et al., 2016), in combination with regularisation mechanisms and self-attention (Saxe et al., 2014; Miyato et al., 2018; Zhang et al., 2019).

Here we aim to improve the adversarial dynamics during training. We focus on the second term in eq. 1 which is at the heart of the min-max game, with adversarial losses for $D$ and $G$, which can be written as

$$L(z) = [L_D(z), L_G(z)]^T = [f(z), -f(z)]^T \tag{2}$$

Computing the gradients with respect to $\theta_D$ and $\theta_G$ obtains the following gradient, which *cannot* be expressed as the gradient of any single function (Balduzzi et al., 2018):

$$g = \left[ \frac{\partial L_D(z)}{\partial \theta_D}, \frac{\partial L_G(z)}{\partial \theta_G} \right]^T = \left[ \frac{\partial f(z)}{\partial \theta_D}, -\frac{\partial f(z)}{\partial \theta_G} \right]^T \tag{3}$$

The fact that $g$ is not the gradient of a function implies that gradient updates in GANs can exhibit cycling behaviour which can slow down or prevent convergence. In Balduzzi et al. (2018), vector fields of this form are referred to as the *simultaneous gradient*. Although many GAN models use alternating update rules (e.g., Goodfellow et al. 2014; Brock et al. 2018), following the gradient with respect to $\theta_D$ and $\theta_G$ alternatively in each step, they can still suffer from cycling, so we use the simpler simultaneous gradient (eq. 3) for our analysis.

## 2.3 LATENT OPTIMISED GANS

Inspired by compressed sensing (Candes et al., 2006; Donoho, 2006), Wu et al. (2019) introduced latent optimisation for GANs. We call this type of model latent-optimised GANs (LOGAN). Latent optimization has been shown to improve the stability of training as well as the final performance for medium-sized models such as DCGANs and Spectral Normalised GANs (Radford et al., 2015; Miyato et al., 2018). Latent optimisation exploits knowledge from $D$ to guide updates of $z$. Intuitively, the gradient $\nabla_z f(z) = \frac{\partial f(z)}{\partial z}$ points in the direction that satisfies the discriminator $D$, which implies better samples. Therefore, instead of using the randomly sampled $z \sim p(z)$, Wu et al. (2019) uses the optimised latent

$$\Delta z = \alpha \frac{\partial f(z)}{\partial z} \qquad z' = z + \Delta z \tag{4}$$

in eq. 1 for training [1]. The general algorithm is summarised in Algorithm 1 and illustrated in Figure 3 **a**. We develop the natural gradient descent form of latent update in Section 4.

---

[1] We use a single step of gradient-based optimisation during training, and justify this choice in section 3.

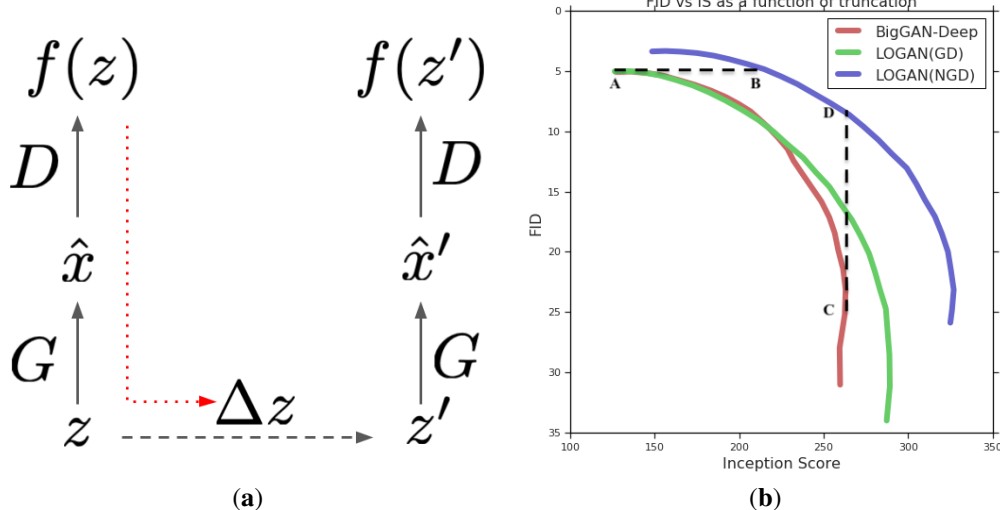

(a)                                                    (b)

Figure 3: (a) Schematic of LOGAN. We first compute a forward pass through $G$ and $D$ with a sampled latent $z$. Then, gradients from the generator loss (dashed red arrow) are used to compute an improved latent, $z'$. After this optimised latent code is used in a second forward pass, we compute gradients of the discriminator back through the latent optimisation into the model parameters $\theta_D$, $\theta_G$. These gradients are used to update the model. (b) Truncation curves illustrate the FID/IS trade-off for each model by altering the range of the noise source $p(z)$. GD: gradient descent. NGD: natural gradient descent. Points A, B, C, D correspond to samples shown in Figure 1 and 2.

---

**Algorithm 1** Latent Optimised GANs with Automatic Differentiation

---

**Input:** data distribution $p(x)$, latent distribution $p(z)$, $D\left(\cdot;\theta_D\right)$, $G\left(\cdot;\theta_G\right)$, learning rate $\alpha$, batch size $N$
**repeat**
    Initialise discriminator and generator parameters $\theta_D$, $\theta_G$
    **for** $i = 1$ **to** $N$ **do**
        Sample $z \sim p(z)$, $x \sim p(x)$
        Compute the gradient $\frac{\partial D(G(z))}{\partial z}$ and use it to obtain $\Delta z$ from eq. 4 (GD) or eq. 12 (NGD)
        Optimise the latent $z' \leftarrow [z + \Delta z]$, $[\cdot]$ indicates clipping the value between $-1$ and $1$
        Compute generator loss $L_G^{(i)} = -D(G(z'))$
        Compute discriminator loss $L_D^{(i)} = D(G(z')) - D(x)$
    **end for**
    Compute batch losses $L_G = \frac{1}{N}\sum_{i=1}^{N} L_G^{(i)}$ and $L_D = \frac{1}{N}\sum_{i=1}^{N} L_D^{(i)}$
    Update $\theta_D$ and $\theta_G$ with the gradients $\frac{\partial L_D}{\partial \theta_D}$, $\frac{\partial L_G}{\partial \theta_G}$
**until** reaches the maximum training steps

---

## 3 ANALYSIS OF THE ALGORITHM

To understand how latent optimisation improves GAN training, we analyse LOGAN as a 2-player differentiable game following Balduzzi et al. (2018); Gemp & Mahadevan (2018); Letcher et al. (2019). The appendix provides a complementary analysis that relates LOGAN to unrolled GANs (Metz et al., 2016) and stochastic approximation (Heusel et al., 2017; Borkar, 1997).

### 3.1 THE SYMPLECTIC GRADIENT ADJUSTMENT (SGA)

An important problem with gradient-based optimization in GANs is that the vector-field generated by the losses of the discriminator and generator is not a gradient vector field. It follows that gradient descent is not guaranteed to find a local optimum and can cycle, which can slow down convergence or lead to phenomena like mode collapse and mode hopping. Balduzzi et al. (2018); Gemp &

Mahadevan (2018) proposed Symplectic Gradient Adjustment (SGA) to improve the dynamics of gradient-based methods in adversarial games. For a game with gradient $g$ (eq. 3), define the Hessian as the second order derivatives with respect to the parameters, $H = \nabla_\theta g$. SGA uses the adjusted gradient

$$g^* = g + \lambda A^T g \quad \text{where } \lambda \text{ is a positive constant} \tag{5}$$

and $A = \frac{1}{2}(H - H^T)$ is the anti-symmetric component of the Hessian. Applying SGA to GANs yields the adjusted updates (see Appendix B.1 for details):

$$g^* = \left[ \frac{\partial f(z)}{\partial \theta_D} + \lambda \left( \frac{\partial^2 f(z)}{\partial \theta_G \, \partial \theta_D} \right)^T \frac{\partial f(z)}{\partial \theta_G}, \quad -\frac{\partial f(z)}{\partial \theta_G} + \lambda \left( \frac{\partial^2 f(z)}{\partial \theta_D \, \partial \theta_G} \right)^T \frac{\partial f(z)}{\partial \theta_D} \right]^T \tag{6}$$

Compared with $g$ in eq. 3, the adjusted gradient $g^*$ has second-order terms reflecting the interactions between $D$ and $G$. SGA has been shown to significantly improve GAN training in basic examples (Balduzzi et al., 2018), allowing faster and more robust convergence to stable fixed points (local Nash equilibria). Unfortunately, SGA is expensive to scale because computing the second-order derivatives with respect to all parameters is expensive.

Explicitly computing the gradients for the discriminator and generator at $z'$ after one step of latent optimisation (eq. 4) obtains

$$\left[ \frac{dL_D}{d\theta_D}, \frac{dL_G}{d\theta_G} \right]^T = \left[ \frac{\partial f(z')}{\partial \theta_D} + \left( \frac{\partial \Delta z}{\partial \theta_D} \right)^T \frac{\partial f(z')}{\partial \Delta z}, \quad -\frac{\partial f(z')}{\partial \theta_G} - \left( \frac{\partial \Delta z}{\partial \theta_G} \right)^T \frac{\partial f(z')}{\partial \Delta z} \right]^T \tag{7}$$

$$= \left[ \frac{\partial f(z')}{\partial \theta_D} + \alpha \left( \frac{\partial^2 f(z)}{\partial z \partial \theta_D} \right)^T \frac{\partial f(z')}{\partial z'}, -\frac{\partial f(z')}{\partial \theta_G} - \alpha \left( \frac{\partial^2 f(z)}{\partial z \partial \theta_G} \right)^T \frac{\partial f(z')}{\partial z'} \right]^T \tag{8}$$

In both equations, the first terms represent how $f(z')$ depends on the parameters directly and the second terms represent how $f(z')$ depends on the parameters via the optimised latent source. For the second equality, we substitute $\Delta z = \alpha \frac{\partial f(z)}{\partial z}$ as the gradient-based update of $z$ and use $\frac{\partial f(z')}{\partial \Delta z} = \frac{\partial f(z')}{\partial z'}$. The original GAN's gradient (eq. 3) does not include any second-order term, since $\Delta z = 0$ without latent optimisation. In LOGAN, these extra terms are computed by automatic differentiation when back-propagating through the latent optimisation process (see Algorithm 1).

The SGA updates in eq. 6 and the LOGAN updates in eq. 8 are strikingly similar, suggesting that the latent step used by LOGAN reduces the negative effects of cycling by introducing a symplectic gradient adjustment into the optimization procedure. The role of the latent step can be formalized in terms of a third player, whose goal is to help the generator, see appendix B for details.

Crucially, latent optimisation approximates SGA using only second-order derivatives with respect to the latent $z$ and parameters of the discriminator and generator *separately*. The second order terms involving parameters of both the discriminator and the generator – which are extremely expensive to compute – are not used. For latent $z$'s with dimensions typically used in GANs (e.g., 128–256, orders of magnitude less than the number of parameters), these can be computed efficiently. In short, latent optimisation *efficiently* couples the gradients of the discriminator and generator, as prescribed by SGA, but using the much lower-dimensional latent source $z$ which makes the adjustment scalable.

An important consequence of reducing the rotational aspect of GAN dynamics is that it is possible to use larger step sizes during training which suggests using stronger optimisers to fully take advantage of latent optimisation. Latent optimisation can improve GAN training dynamics further by allowing *larger* single steps $\Delta z$ towards the direction of $\frac{\partial f(z)}{\partial z}$ *without overshooting*.

### 3.2 UNROLLING AND STOCHASTIC APPROXIMATION

Appendix B further explains how LOGAN relates to unrolled GANs (Metz et al., 2016) and stochastic approximation. Our main finding is that latent optimisation accelerates the speed of updating $D$ relative to that of $G$, facilitating convergence according to Heusel et al. (2017) (see also Figure 4 **b**). In particular, the generator requires less update compared with $D$ to achieve the same reduction of loss, because latent optimisation "helps" $G$.

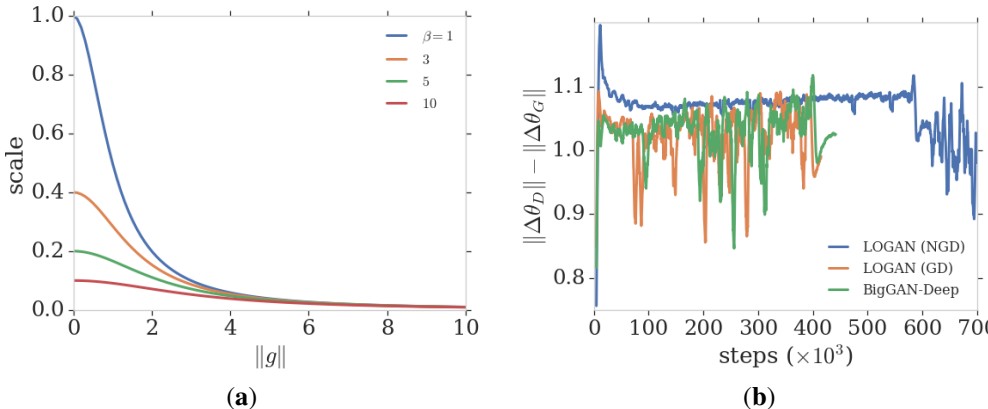

Figure 4: (**a**) Scaling of gradients in natural gradient descent. We use $\beta = 5$ in BigGAN-Deep experiments. (**b**) The update speed of the discriminator relative to the generator shown as the difference $\|\Delta\theta_D\| - \|\Delta\theta_G\|$ after each update step. Lines are smoothed with moving average using window size 20 (in total, there are 3007, 1659 and 1768 data points for each curve). For all curves oscillation strongly after training collapsed.

## 4   LOGAN WITH NATURAL GRADIENT DESCENT

Although our analysis suggests using strong optimisers for optimising $z$, Wu et al. (2019) only used basic gradient descent (GD) with a fixed step-size. This choice limits the size $\Delta z$ can take: in order not to overshoot when the curvature is large, the step size would be too conservative when the curvature is small. We hypothesis that GD is more detrimental for larger models, which have complex loss surfaces with highly varying curvatures. Consistent with this hypothesis, we observed only marginal improvement over the baseline using GD (section 5.3, Table 1, Figure 3 **b**).

In this work, we instead use natural gradient descent (NGD, Amari 1998) for latent optimisation. NGD can be seen as an approximate second-order optimisation method (Pascanu & Bengio, 2013; Martens, 2014), and has been applied successfully in many domains. By using the positive semi-definite (PSD) Gauss-Newton matrix to approximate the (possibly negative definite) Hessian, NGD often works even better than exact second-order methods. NGD is expensive in high dimensional parameter spaces, even with approximations (Martens, 2014). However, we demonstrate it is efficient for latent optimisation, even in very large models.

Given the gradient of $z$, $g = \frac{\partial f(z)}{\partial z}$, NGD computes the update as

$$\Delta z = \alpha \, F^{-1} \, g \tag{9}$$

where the Fisher information matrix $F$ is defined as

$$F = \mathbb{E}_{p(t|z)} \left[ \nabla \ln p(t|z) \, \nabla \ln p(t|z)^T \right] \tag{10}$$

The log-likelihood function $\ln p(t|z)$ typically corresponds to commonly used error functions such as cross entropy loss. This correspondence is not necessary when NGD is interpreted as an approximate second-order method, as has long been used in practice (Martens, 2014). Nevertheless, Appendix C provides a Poisson log-likelihood interpretation for the hinge loss commonly used in GANs (Lim & Ye, 2017; Tran et al., 2017). An important difference between latent optimisation and commonly seen senarios using NGD is that the expectation over the condition ($z$) is absent. Since each $z$ is only responsible for generating one image, it only minimises the loss $L_G(z)$ for this particular instance. Computing per-sample Fisher this way is necessary to approximate SGA (see Appendix B.1 for details).

More specifically, we use the *empirical* Fisher $F'$ with Tikhonov damping, as in TONGA (Roux et al., 2008)

$$F' = g \cdot g^T + \beta \, I \tag{11}$$

$F'$ is cheaper to compute compared with the full Fisher, since $g$ is already available. The *damping factor* $\beta$ regularises the step size, which is important when $F'$ only poorly approximates the Hessian

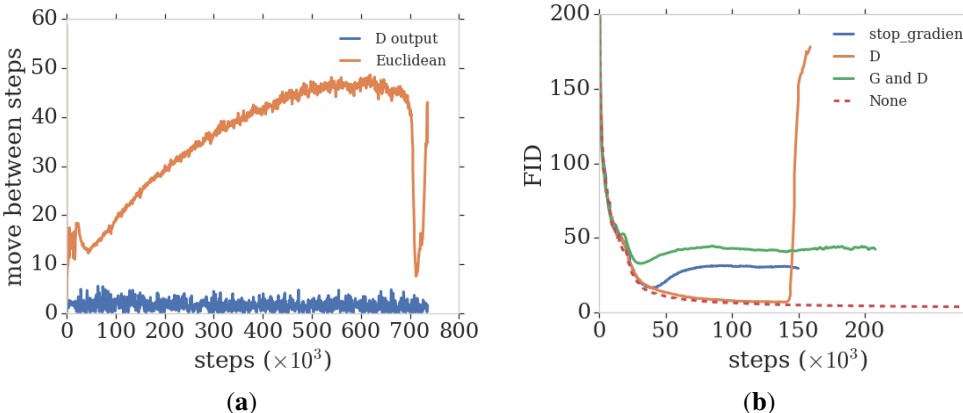

Figure 5: (**a**) The change from $\Delta z$ across training, in $D$'s output space and $z$'s Euclidean space. The distances are normalised by their standard derivations computed from a moving window of size 20 (1007 data points in total). (**b**) Training curves from models with different "`stop_gradient`" operations. For reference, the training curve from an unablated model is plotted as the dashed line. All instances with `stop_gradient` collapsed (FID went up) early in training.

or when the Hessian changes too much across the step. Using the Sherman-Morrison formula, the NGD update can be simplified into the following closed form:

$$\Delta z = \alpha \left( \frac{I}{\beta} - \frac{g\,g^T}{\beta^2 + \beta\,g^T g} \right) g = \frac{\alpha}{\beta} \left( 1 - \frac{\|g\|^2}{\beta + \|g\|^2} \right) g \tag{12}$$

which does not involve any matrix inversion. Thus, NGD adapts the step size according to the curvature estimate $c = \frac{1}{\beta} \left( 1 - \frac{\|g\|^2}{\beta + \|g\|^2} \right)$. Figure 4 **a** illustrates the scaling for different values of $\beta$. NGD automatically smooths the scale of updates by down-scaling the gradients as their norm grows, which also contributes to the smoothed norms of updates (Figure 4 **b**). Since the NGD update remains proportional to $g$, our analysis based on gradient descent in section 3 still holds.

## 5 EXPERIMENTS AND ANALYSIS

We focus on large scale GANs based on BigGAN-deep (Brock et al., 2018) trained on $128 \times 128$ size images from the ImageNet dataset (Deng et al., 2009). In Appendix E, we present results from applying our algorithm on Spectral Normalised GANs trained with CIFAR dataset (Krizhevsky et al., 2009), which obtains state-of-the-art scores on this model.

### 5.1 MODEL CONFIGURATION

We used the standard BigGAN-deep architecture with three minor modifications: 1. We increased the size of the latent source from 128 to 256, to compensate the randomness of the source lost when optimising $z$. 2. We use the uniform distribution $\mathcal{U}(-1, 1)$ instead of the standard normal distribution $\mathcal{N}(0, 1)$ for $p(z)$, to be consistent with the clipping operation (Algorithm 1). 3. We use `leaky ReLU` instead of `ReLU` as the non-linearity for smoother gradient flow for $\frac{\partial f(z)}{\partial z}$.

Consistent with detailed findings in Brock et al. (2018) that these changes have limited effect, our experiment with this baseline model obtains only slightly better scores compared with those in Brock et al. (2018) (Table 1, see also Figure 8). The FID and IS are computed as in Brock et al. (2018), and IS values are computed from checkpoints with the lowest FIDs. The means and standard deviations are computed from 5 models with different random seeds.

To apply latent optimisation, we use a damping factor $\beta = 5.0$ combined with a large step size of $\alpha = 0.9$. As an additional way of damping, we only optimise 50% of $z$'s dimensions. Optimising the entire population of $z$ was unstable in our experiments. Similar to Wu et al. (2019), we found it was helpful to regularise the Euclidean norm of weight-change $\Delta z$, with a regulariser weight of

300.0. All other hyper-parameters, including learning rates and a large batch size of 2048, remain the same as in BigGAN-deep; we did not optimise these hyper-parameters. We call this model LOGAN (NGD).

## 5.2 BASIC RESULTS

Employing the same architecture and number of parameters as the BigGAN-deep baseline, LOGAN (NGD) achieved better FID and IS (Table 1). As observed by Brock et al. (2018), BigGAN training always eventually collapsed. Training with LOGAN also collapsed, perhaps due to higher-order dynamics beyond the scope we have analysed, but took significantly longer (600k steps versus 300k steps with BigGAN-deep).

During training, LOGAN was $2 - 3$ times slower per step compared with BigGAN-deep because of the additional forward and backward pass. We found that optimising $z$ during evaluation did not improve sample scores (even up to 10 steps), so we do not optimise $z$ for evaluation. Therefore, LOGAN has the same evaluation cost as original BigGAN-deep. To help understand this behaviour, we plot the change from $\Delta z$ during training in Figure 5 **a**. Although the movement in Euclidean space $\|\Delta z\|$ grew until training collapsed, the movement in $D$'s output space, measured as $\|f(z + \Delta z) - f(z)\|$, remained unchanged (see Appendix D for details). As shown in our analysis, optimising $z$ improves the training dynamics, so LOGANs work well after training without requiring latent optimisation.

## 5.3 ABLATION STUDIES

We verify our theoretical analysis in section 3 by examining key components of Algorithm 1 via ablation studies. First, we experimented with using basic GD to optimising $z$, as in Wu et al. (2019), and call this model LOGAN (GD). A smaller step size of $\alpha = 0.0001$ was required; larger values were unstable and led to premature collapse of training. As shown in Table 1, the scores from LOGAN (GD) were worse than LOGAN (NGD) and similar to the baseline model.

We then evaluate the effects of removing those terms depending on $\frac{\partial f(z)}{\partial z}$ in eq. 8, which are not in the ordinary gradient (eq. 3). Since these terms were computed when back-propagating through the latent optimisation procedure, we removed them by selectively blocking back-propagation with "`stop_gradient`" operations (e.g., in TensorFlow Abadi et al. 2016). Figure 5 **b** shows the change of FIDs for the three models corresponding to removing $\left(\frac{\partial \Delta z}{\partial \theta_G}\right)^T \frac{\partial f(z')}{\partial z'}$, removing $\left(\frac{\partial \Delta z}{\partial \theta_D}\right)^T \frac{\partial f(z')}{\partial z'}$ and removing both terms. As predicted by our analysis (section 3), both terms help stabilise training; training diverged early for all three ablations.

## 5.4 TRUNCATION AND SAMPLES

Truncation is a technique introduced by Brock et al. (2018) to illustrate the trade-off between the FID and IS in a trained model. For a model trained with $z \sim p(z)$ from a source distribution symmetric around 0, such as the standard normal distribution $\mathcal{N}(0, 1)$ and the uniform distribution $\mathcal{U}(-1, 1)$, down-scaling (truncating) the source $\bar{z} = s \cdot z$ with $0 \leq s \leq 1$ gives samples with higher visual quality but reduced diversity. This observation is quantified as higher IS and lower FID when evaluating samples from truncated distributions.

Figure 3 **b** plots the truncation curves for the baseline BigGAN-deep model, LOGAN (GD) and LOGAN (NGD), obtained by varying the truncation (value of $s$) from $1.0$ (no truncation, upper-left ends of the curves) to $0.02$ (extreme truncation, bottom-right ends). Each curve shows the trade-off between FID and IS for an individual model; curves towards the upper-right corner indicate better overall sample quality. The relative positions of curves in Figure 3 (**b**) shows LOGAN (NGD) has the best sample quality. Interestingly, although LOGAN (GD) and the baseline model have similar scores without truncation (upper-left ends of the curves, see also Table 1), LOGAN (GD) was better behaved with increasing truncation, suggesting LOGAN (GD) still converged to a better equilibrium. For further reference, we plot truncation curves from additional baseline models in Figure 8.

Figure 1 and Figure 2 show samples from chosen points on the truncation curves. In the high IS domain, C and D on the truncation curves both have similarly high IS of near 260. Samples from batches with such high IS have almost photo-realistic image quality. Figure 1 show that while the baseline model produced nearly uniform samples, LOGAN (NGD) could still generate highly diverse samples. On the other hand, A and B from Figure 3 **b** have similarly low FID of near 5, indicating high sample diversity. Samples in Figure 2 **b** show higher quality compared with those in **a** (e.g., the interfaces between the elephants and ground, the contours around the pandas).

## 6 CONCLUSION

In this work we present the LOGAN model which significantly improves the state-of-the-art on large scale GAN training for image generation by online optimising the latent source $z$. Our results illustrate improvements in quantitative evaluation and samples with higher quality and diversity. Moreover, our analysis suggests that LOGAN fundamentally improves adversarial training dynamics. We therefore expect our method to be useful in other tasks that involve adversarial training, including representation learning and inference (Donahue et al., 2017; Dumoulin et al., 2017), text generation (Zhang et al., 2019), style learning (Zhu et al., 2017; Karras et al., 2019), audio generation (Donahue et al., 2018) and video generation (Vondrick et al., 2016; Clark et al., 2019).

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

## A  ADDITIONAL SAMPLES AND RESULTS

Figure 6 and 7 provide additional samples, organised similarly as in Figure 1 and 2. Figure 8 shows additional truncation curves.

## B  DETAILED ANALYSIS OF LATENT OPTIMISATION

In this section we present three complementary analyses of LOGAN. In particular, we show how the algorithm brings together ideas from symplectic gradient adjustment, unrolled GANs and stochastic approximation with two time scales.

### B.1  APPROXIMATE SYMPLECTIC GRADIENT ADJUSTMENT

To analyse LOGAN as a differentiable game we treat the latent step $\Delta z$ as adding a *third player* to the original game played by the discriminator and generator. The third player's parameter, $\Delta z$, is optimised online for each $z \sim p(z)$. Together the three players (latent player, discriminator, and generator) have losses *averaged over a batch of samples*:

$$L = [\eta\, L_G, L_D, L_G]^T \tag{13}$$

where $\eta = \frac{1}{N}$ ($N$ is the batch size) reflects the fact that each $\Delta z$ is only optimised for a single sample $z$, so its contribution to the total loss across a batch is small compared with $\theta_D$ and $\theta_G$ which are directly optimised for batch losses. This choice of $\eta$ is essential for the following derivation, and has important practical implication. It means that the per-sample loss $L_G(z')$, instead of the loss summed over a batch $\sum_{n=1}^{N} L_G(z'_n)$, should be the only loss function guiding latent optimisation. Therefore, when using natural gradient descent (Section 4), the Fisher information matrix should only be computed using the current sample $z$.

The resulting simultaneous gradient is

$$g = \left[\eta\, \frac{\partial L_G(z')}{\partial \Delta z}, \frac{\partial L_D(z')}{\partial \theta_D}, \frac{\partial L_G(z')}{\partial \theta_G}\right]^T = \left[-\eta\, \frac{\partial f(z')}{\partial \Delta z}, \frac{\partial f(z')}{\partial \theta_D}, -\frac{\partial f(z')}{\partial \theta_G}\right]^T \tag{14}$$

Following Balduzzi et al. (2018), we can write the Hessian of the game as:

$$H = \begin{bmatrix} -\eta\, \frac{\partial^2 f(z')}{\partial \Delta z^2} & -\eta\, \frac{\partial^2 f(z')}{\partial \Delta z \partial \theta_D} & -\eta\, \frac{\partial^2 f(z')}{\partial \Delta z \partial \theta_G} \\ \frac{\partial^2 f(z')}{\partial \theta_D \partial \Delta z} & \frac{\partial^2 f(z')}{\partial \theta_D^2} & \frac{\partial^2 f(z')}{\partial \theta_D \partial \theta_G} \\ -\frac{\partial^2 f(z')}{\partial \theta_G \partial \Delta z} & -\frac{\partial^2 f(z')}{\partial \theta_G \partial \theta_D} & -\frac{\partial^2 f(z')}{\partial \theta_G^2} \end{bmatrix} \tag{15}$$

The presence of a non-zero anti-symmetric component in the Hessian

$$
A = \frac{1}{2}(H - H^T) = \begin{bmatrix} 0 & -\frac{1+\eta}{2} \frac{\partial^2 f(z')}{\partial \Delta z \partial \theta_D} & \frac{1-\eta}{2} \frac{\partial^2 f(z')}{\partial \Delta z \partial \theta_G} \\ \frac{1+\eta}{2} \frac{\partial^2 f(z')}{\partial \theta_D \partial \Delta z} & 0 & \frac{\partial^2 f(z')}{\partial \theta_D \partial \theta_G} \\ -\frac{1-\eta}{2} \frac{\partial^2 f(z')}{\partial \theta_G \partial \Delta z} & -\frac{\partial^2 f(z')}{\partial \theta_G \partial \theta_D} & 0 \end{bmatrix} \tag{16}
$$

implies the dynamics have a rotational component which can cause cycling or slow down convergence. Since $\eta \ll 1$ for typical batch sizes (e.g., $\frac{1}{64}$ for DCGAN and $\frac{1}{2048}$ for BigGAN-deep), we abbreviate $\gamma = \frac{1+\eta}{2} \approx \frac{1-\eta}{2}$ to simplify notations.

Symplectic gradient adjustment (SGA) counteracts the rotational force by adding an adjustment term to the gradient to obtain $g^* \leftarrow g + \lambda A^T g$, which for the discriminator and generator has the form:

$$
g_D^* = \frac{\partial f(z')}{\partial \theta_D} + \lambda\gamma \left( \frac{\partial^2 f(z')}{\partial \Delta z \partial \theta_D} \right)^T \frac{\partial f(z')}{\partial \Delta z} + \lambda \left( \frac{\partial^2 f(z')}{\partial \theta_G \, \partial \theta_D} \right)^T \frac{\partial f(z')}{\partial \theta_G} \tag{17}
$$

$$
g_G^* = -\frac{\partial f(z')}{\partial \theta_G} - \lambda\gamma \left( \frac{\partial^2 f(z')}{\partial \Delta z \partial \theta_G} \right)^T \frac{\partial f(z')}{\partial \Delta z} + \lambda \left( \frac{\partial f(z')}{\partial \theta_D \, \partial \theta_G} \right)^T \frac{\partial f(z')}{\partial \theta_D} \tag{18}
$$

The gradient with respect to $\Delta z$ is ignored since the convergence of training only depends on $\theta_D$ and $\theta_G$.

If we drop the last terms in eq.17 and 18, which are expensive to compute for large models with high-dimensional $\theta_D$ and $\theta_G$, and use $\frac{\partial f(z')}{\partial \Delta z} = \frac{\partial f(z')}{\partial z'}$, the adjusted updates can be rewritten as

$$
g_D^* \approx \frac{\partial f(z')}{\partial \theta_D} + \lambda\gamma \left( \frac{\partial^2 f(z')}{\partial z' \partial \theta_D} \right)^T \frac{\partial f(z')}{\partial z'} \tag{19}
$$

$$
g_G^* \approx -\frac{\partial f(z')}{\partial \theta_G} - \lambda\gamma \left( \frac{\partial^2 f(z')}{\partial z' \partial \theta_G} \right)^T \frac{\partial f(z')}{\partial z'} \tag{20}
$$

Because of the third player, there are still the terms depend on $\frac{\partial f(z')}{\partial z'}$ to adjust the gradients. Efficiently computing $\frac{\partial^2 f(z')}{\partial z' \partial \theta_D}$ and $\frac{\partial^2 f(z')}{\partial z' \partial \theta_D}$ is non-trivial (e.g., Pearlmutter 1994). However, if we introduce the local approximation

$$
\frac{\partial^2 f(z')}{\partial z' \partial \theta_D} \approx \frac{\partial^2 f(z)}{\partial z \partial \theta_D} \qquad \frac{\partial^2 f(z')}{\partial z' \partial \theta_D} \approx \frac{\partial^2 f(z)}{\partial z \partial \theta_D} \tag{21}
$$

then the adjusted gradient becomes identical to 8 from latent optimisation.

In other words, automatic differentiation by commonly used machine learning packages can compute the adjusted gradient for $\theta_D$ and $\theta_G$ when back-propagating through the latent optimisation process. Despite the approximation involved in this analysis, both our experiments in section 5 and the results from Wu et al. (2019) verified that latent optimisation can significantly improve GAN training.

## B.2 RELATION WITH UNROLLED GANS

Latent optimisation can be seen as unrolling GANs (Metz et al., 2016) in the space of the latent, rather than the parameters. Unrolling in the latent space has the advantages that:

1. LOGAN is more scalable than Unrolled GANs because it avoids second-order derivatives over a potentially very large number of parameters.
2. While unrolling the update of $D$ only affects the parameters of $G$ (as in Metz et al. 2016), latent optimisation effects both $D$ and $G$ as shown in eq. 8.

We next formally present this connection by showing that SGA can be seen as approximating Unrolled GANs (Metz et al., 2016). For the update $\theta_D' = \theta_D + \Delta\theta_D$, we have the Taylor expansion approximation at $\theta_D$:

$$
f(z; \theta_D + \Delta\theta_D, \theta_G) \approx f(z; \theta_D, \theta_G) + \left( \frac{\partial f(z; \theta_D, \theta_G)}{\partial \theta_D} \right)^T \Delta\theta_D \tag{22}
$$

Substitute $\Delta\theta_D = -\alpha\frac{\partial f(z;\theta_D,\theta_G)}{\partial\theta_D}$, and take the derivatives with respect to $\theta_G$ on both sides:

$$\frac{\partial f(z;\theta_D + \Delta\theta_D, \theta_G)}{\partial\theta_G} \approx \frac{\partial f(z;\theta_D,\theta_G)}{\partial\theta_G} - 2\alpha\left(\frac{\partial^2 f(z;\theta_D,\theta_G)}{\partial\theta_D\partial\theta_G}\right)^T \frac{\partial f(z;\theta_D,\theta_G)}{\partial\theta_D} \tag{23}$$

which is the same as eq. 18 (taking the negative sign). Compared with the exact gradient from the unroll:

$$\frac{\partial f(z;\theta_D + \Delta\theta_D, \theta_G)}{\partial\theta_G} = \frac{\partial f(z;\theta'_D,\theta_G)}{\partial\theta_G} - 2\alpha\left(\frac{\partial^2 f(z;\theta_D,\theta_G)}{\partial\theta_D\partial\theta_G}\right)^T \frac{\partial f(z;\theta'_D,\theta_G)}{\partial(\theta'_D)} \tag{24}$$

The approximation in eq. 23 comes from using $\frac{\partial f(z;\theta_D,\theta_G)}{\partial\theta_D} \approx \frac{\partial f(z;\theta'_D,\theta_G)}{\partial\theta'_D}$ and $\frac{\partial f(z;\theta_D,\theta_G)}{\partial\theta_G} \approx \frac{\partial f(z;\theta'_D,\theta_G)}{\partial\theta_G}$ as a result of the linear approximation.

At this point, unrolling $D$ update only affects $\theta_D$. Although it is expensive to unroll both $D$ and $G$, in principle, we can unroll $G$ update and compute the gradient of $\theta_D$ similarly using $\Delta\theta_G = \alpha\frac{\partial f(z;\theta_D,\theta_G)}{\partial\theta_G}$:

$$\frac{\partial f(z;\theta_D, \theta_G + \Delta\theta_G)}{\partial\theta_D} \approx \frac{\partial f(z;\theta_D,\theta_G)}{\partial\theta_D} + 2\alpha\left(\frac{\partial^2 f(z;\theta_D,\theta_G)}{\partial\theta_G\partial\theta_D}\right)^T \frac{\partial f(z;\theta_D,\theta_G)}{\partial\theta_G} \tag{25}$$

which gives us the same update rule as SGA (eq. 17). This correspondence based on first order Taylor expansion is unsurprising, as SGA is based on linearising the adversarial dynamics (Balduzzi et al., 2018).

## B.3 Stochastic Approximation with Two Time Scales

Heusel et al. (2017) used the theory of stochastic approximation to analyse GAN training. Viewing the training process as stochastic approximation with two time scales (Borkar, 1997; Konda & Borkar, 1999), they suggest that the update of $D$ should be fast enough compared with that of $G$. Under mild assumptions, Heusel et al. (2017) proved that such two time-scale update converges to local Nash equilibrium. Their analysis follows the idea of $(\tau, \delta)$ perturbation (Hirsch, 1989), where the slow updates ($G$) is interpreted as a small perturbation over the ODE describing the fast update ($D$). Importantly, the size of perturbation $\delta$ is measured in the magnitude of parameter change, which is affected by both the learning rate and gradients.

Here we show that LOGAN accelerates discriminator updates and slows down generator updates, thus helping the convergence of discriminator according to Heusel et al. (2017). We start from analysing the change of $\theta_G$. We assume that, without LO, it takes $\Delta\theta_G = \theta'_G - \theta_G$ to make a small constant amount of reduction in loss $L_G$:

$$\rho = -f(z;\theta_D, \theta_G + \Delta\theta_G) + f(z;\theta_D,\theta_G) \tag{26}$$

Now using the optimised $z' = z + \Delta z$, we assess the change $\delta\theta_G$ required to achieve the same amount of reduction:

$$\rho = -f(z + \Delta z;\theta_D, \theta_G + \delta\theta_G) + f(z;\theta_D,\theta_G) \tag{27}$$

Intuitively, when $z$ "helps" $\theta_G$ to achieve the same goal of increasing $f(z;\theta_D,\theta_G)$ by $\rho$, the responsible of $\theta_G$ becomes smaller, so it does not need to change as much as $\Delta\theta_G$, thus $\|\delta\theta_G\| < \|\Delta\theta_G\|$.

Formally, $f(z;\theta_D,\theta_G)$ and $f(z + \Delta;\theta_D, \theta_G + \delta\theta_G)$ have the following Taylor expansions around $z$ and $\theta_G$:

$$f(z;\theta_d, \theta_G + \delta\theta_G) = f(z;\theta_D,\theta_G) + \left(\frac{\partial f(z;\theta_D,\theta_G)}{\partial\theta_G}\right)^T \Delta\theta_G + \epsilon(\Delta\theta_G) \tag{28}$$

$$f(z + \Delta z;\theta_d, \theta_G + \delta\theta_G) = f(z;\theta_D,\theta_G) + \left(\frac{\partial f(z;\theta_D,\theta_G)}{\partial z}\right)^T$$
$$\Delta z + \left(\frac{\partial f(z + \Delta z;\theta_D,\theta_G)}{\partial\theta_G}\right)^T \delta\theta_G + \epsilon(\Delta z, \delta\theta_G) \tag{29}$$

Where $\epsilon(\cdot)$'s are higher order terms of the increments. Using the assumption of eq. 26 and 27, we can combine eq. 28 and 29:

$$\left(\frac{\partial f(z;\theta_D,\theta_G)}{\partial \theta_G}\right)^T \Delta\theta_G = \left(\frac{\partial f(z;\theta_D,\theta_G)}{\partial z}\right)^T \Delta z + \left(\frac{\partial f(z+\Delta z;\theta_D,\theta_G)}{\partial \theta_G}\right)^T \delta\theta_G + \epsilon \quad (30)$$

where $\epsilon = \epsilon(\Delta z, \delta\theta_G) - \epsilon(\Delta\theta_G)$. Since $\Delta z \propto \frac{\partial f(z;\theta_D,\theta_G)}{\partial z}$ in gradient descent (eq. 3),

$$\frac{\partial f(z;\theta_D,\theta_G)}{\partial z}\Delta z > 0 \quad (31)$$

Therefore, we have the inequality

$$\left(\frac{\partial f(z;\theta_D,\theta_G)}{\partial \theta_G}\right)^T \Delta\theta_G < \left(\frac{\partial f(z+\Delta z;\theta_D,\theta_G)}{\partial \theta_G}\right)^T \delta\theta_G + \epsilon \quad (32)$$

If we further assume $\Delta\theta_G$ and $\delta\theta_G$ are obtained from stochastic gradient descent with identical learning rate,

$$\Delta\theta_G = \alpha \frac{\partial f(z;\theta_D,\theta_G)}{\partial \theta_G} \qquad \delta\theta_G = \alpha \frac{\partial f(z;\theta_D,\theta_G)}{\partial \theta_G} \quad (33)$$

substituting eq. 33 into eq. 32 gives

$$\|\Delta\theta_G\| < \|\delta\theta_G\| + \epsilon \quad (34)$$

The same analysis applies to the discriminator. The similar intuition is that it takes the discriminator additional effort to compensate the exploitation from the optimised $z'$. We then obtain

$$\left(\frac{\partial f(z;\theta_D,\theta_G)}{\partial \theta_D}\right)^T \Delta\theta_D = \left(\frac{\partial f(z;\theta_D,\theta_G)}{\partial z}\right)^T \Delta z + \left(\frac{\partial f(z+\Delta z;\theta_D,\theta_G)}{\partial \theta_D}\right)^T \delta\theta_D + \epsilon \quad (35)$$

However, since the adversarial loss $L_D = -L_G$, we have $\Delta\theta_D = -\alpha \frac{\partial f(z;\theta_D,\theta_G)}{\partial \theta_D}$ and $\delta\theta_D = -\alpha \frac{\partial f(z;\theta_D,\theta_G)}{\partial \theta_D}$ taking the opposite signs of eq.33. For sufficiently small $\Delta z$, $\Delta\theta_G$ and $\delta\theta_G$, $\epsilon$ is close to zero, so $\|\Delta\theta_D\| < \|\delta\theta_D\|$ under our assumptions of small $\Delta z$, $\Delta\theta_G$ and $\delta\theta_G$.

Importantly, the bigger the product $\frac{\partial f(z)}{\partial z}\Delta z$ is, the more robust the inequality is to the error from $\epsilon$. Moreover, bigger step increases the speed gap between updating D and G, further facilitating convergence according to Heusel et al. (2017). Overall, our analysis suggests:

1. More than one gradient descent step may not be helpful, since $\Delta z$ from multiple GD steps may deviate from the direction of $\frac{\partial f(z)}{\partial z}$.

2. Large step of $\Delta z$ is more helpful in facilitating convergence by widening the gap between D and G updates (Heusel et al., 2017).

3. However, the step of $\Delta z$ cannot be too large. In addition to the linear approximation we used throughout our analysis, the approximate SGA breaks down when eq.21 is strongly violated when "overshoot" brings the gradients at $\frac{\partial f(z')}{\partial z'}$ to the opposite sign of $\frac{\partial f(z)}{\partial z}$.

## C  POISSON LIKELIHOOD FROM HINGE LOSS

Here we provide a probabilistic interpretation of the hinge loss for the generator, which leads naturally to the scenario of a family of discriminators. Although this interpretation is not necessary for our current algorithm, it may provides useful guidance for incorporating multiple discriminators.

We introduce the label $t = 1$ for real data and $t = 0$ fake samples. This section shows that the generator hinge loss

$$L_G = -D\left(G(z)\right) \quad (36)$$

can be interpreted as a negative log-likelihood function:

$$L_G = -\ln p(t = 1; D, G(z)) \quad (37)$$

Here $p(t = 1; z, D, G)$ is the probability that the generated image $G(z)$ can fool the discriminator $D$.

The original GAN's discriminator can be interpreted as outputting a Bernoulli distribution $p(t; \beta_G) = \beta_G^t \cdot (1 - \beta_G)^{1-t}$. In this case, if we parameterise $\beta_G = D(G(z))$, the generator loss is the negative log-likelihood

$$- \ln P\big(t = 1; D, G(z)\big) = - \ln p(t = 1; \beta_G) = - \ln \beta_G = - \ln D(G(z)) \tag{38}$$

Bernoulli, however, is not the only valid choice as the discriminator's output distribution. Instead of sampling "1" or "0", we assume that there are *many* identical discriminators that can independently vote to reject an input sample as fake. The number of votes $k$ in a given interval can be described by a Poisson distribution with parameter $\lambda$ with the following PMF:

$$p(k; \lambda) = \frac{\lambda^k e^{-\lambda}}{k!} \tag{39}$$

The probability that a generated image can fool *all* the discriminators is the probability of $G(z)$ receiving no vote for rejection

$$p(k = 0; \lambda) = e^{-\lambda} \tag{40}$$

Therefore, we have the following negative log-likelihood as the generator loss if we parameterise $\lambda = -D(G(z))$:

$$- \ln p\big(k = 0; D, G(z)\big) = - \ln p(k = 0; \lambda) = -D(G(z)) \tag{41}$$

This interpretation has a caveat that when $D(G(z)) > 0$ the Poisson distribution is not well defined. However, in general the discriminator's hinge loss

$$L_D = - \min\big(0, -1 + D(x)\big) - \min\big(0, -1 - D(G(z))\big) \tag{42}$$

pushes $D(G(z)) < 0$ via training.

## D  DETAILS IN COMPUTING DISTANCES IN FIGURE 5 A

For a temporal sequence $x_1, x_2, \ldots, x_T$ (changes of $z$ or $f(z)$ at each training step in this paper), to normalise its variance while accounting for the non-stationarity, we process it as follows. We first compute the moving average and standard deviation over a window of size $N$:

$$\mu_t = \frac{1}{N} \sum_{u=t}^{t+N-1} x_u \tag{43}$$

$$\sigma_t = \sqrt{\frac{1}{N-1} \sum_{u=t}^{t+N-1} (x_u - \mu_u)^2} \tag{44}$$

Then normalise the sequence as:

$$\bar{x}_t = \frac{x_t}{\sigma_t} \tag{45}$$

The result in Figure 5 **a** is robust to the choice of window size. Our experiments with $N$ from 10 to 50 yielded visually similar plots.

## E  EXPERIMENTS WITH DCGAN AND CIFAR

To test if latent optimisation works with models at more moderate scales, we applied it on SN-GANs (Miyato et al., 2018). Although our experiments on this model are less thorough than in the main paper with BigGAN-deep, we hope to provide basic guidelines for researchers interested in applying latent optimisation on smaller models.

The experiments follows the same basic setup and hyper-parameter settings as the CS-GAN in Wu et al. (2019). There is no class conditioning in this model. For NGD, we found a smaller damping

factor $\beta = 0.1$, a $\|z\|$ regulariser weight of 3.0 (the same as in Wu et al. 2019), combined with optimising 70% of the latent source (instead of 50% for BigGAN-deep) worked best for SN-GANs.

In addition, we found running extra latent optimisation steps benefited evaluation, so we use ten steps of latent optimisation in evaluation for results in this section, although the models were still trained with a single optimisation step. We reckon that smaller models might not be "over-parametrised" enough to fully amortise the computation from optimising $z$, which can then further exploit the architecture in evaluation time. On the other hand, the overhead from running multiple iterations of latent optimisation is relatively small at this scale. We aim to further investigate this difference in future studies.

Table 2 shows the FID and IS alongside SN-GAN and CS-CAN which used the same architecture. Here we observe similarly significant improvement over the baseline SN-GAN model, with an improvement of 16.8% in IS and 39.6% in FID. Figure 9 shows random samples from these two models. Overall, samples from LOGAN (NGD) have higher contrasts and sharper contours.

Table 2: Comparison of Scores. The first and second columns are reproduced from Miyato et al. (2018) and Wu et al. (2019) respectively. We report the Inception Score (IS, higher is better, Salimans et al. 2016) and Fréchet Inception Distance (FID, lower is better, Heusel et al. 2017).

|  | SN-GAN | CS-GAN | LOGAN (NGD) |
|---|---|---|---|
| FID | 29.3 | $23.1 \pm 0.5$ | $\mathbf{17.7 \pm 0.4}$ |
| IS | $7.42 \pm 0.08$ | $7.80 \pm 0.05$ | $\mathbf{8.67 \pm 0.05}$ |

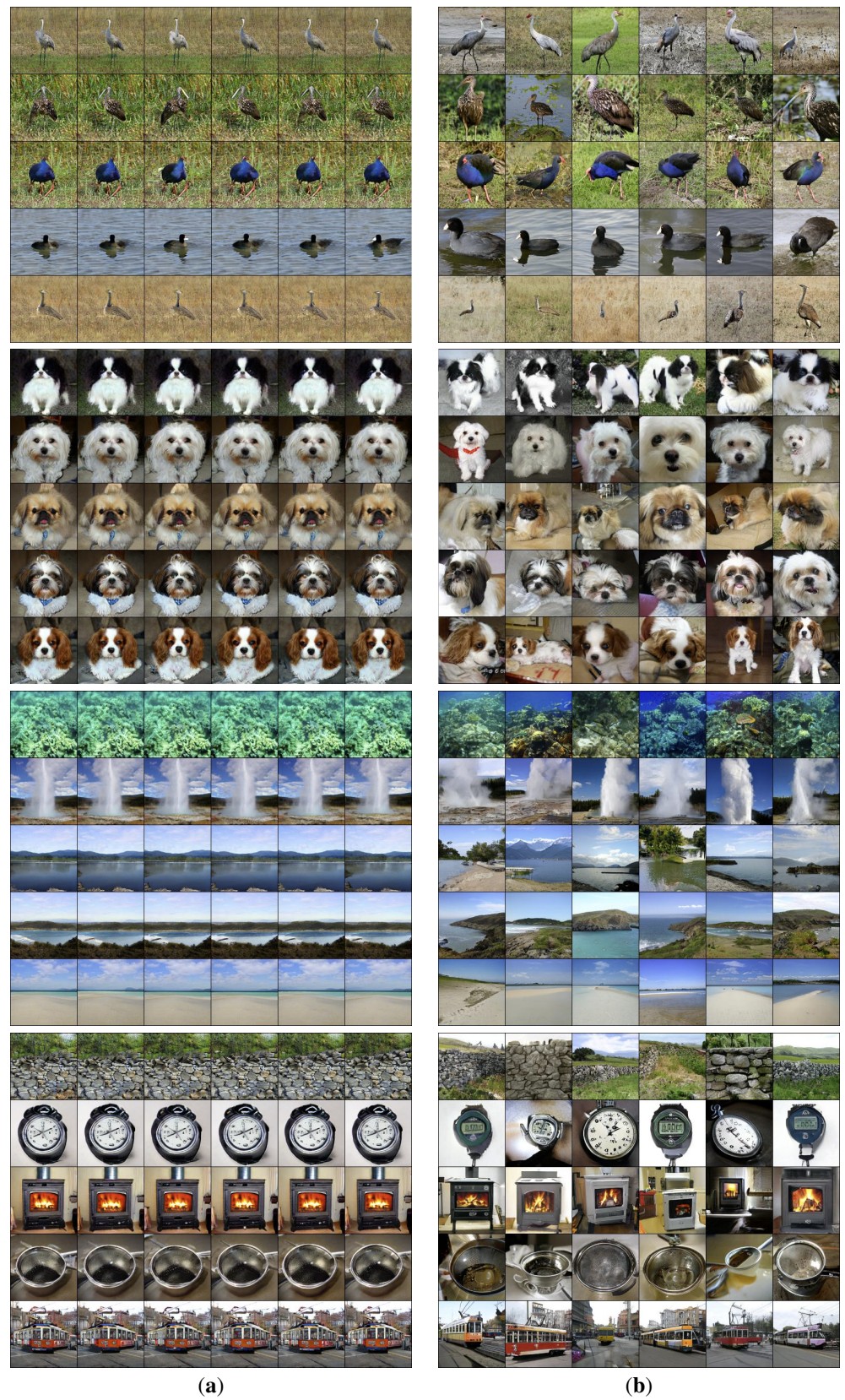

Figure 6: Samples from BigGAN-deep (**a**) and LOGAN (**b**) with the similarly high inception scores. Samples from the two panels were draw from truncations correspond to points C, D in figure 3 **b** respectively. (FID/IS: (**a**) 27.97/259.4, (**b**) 8.19/259.9)

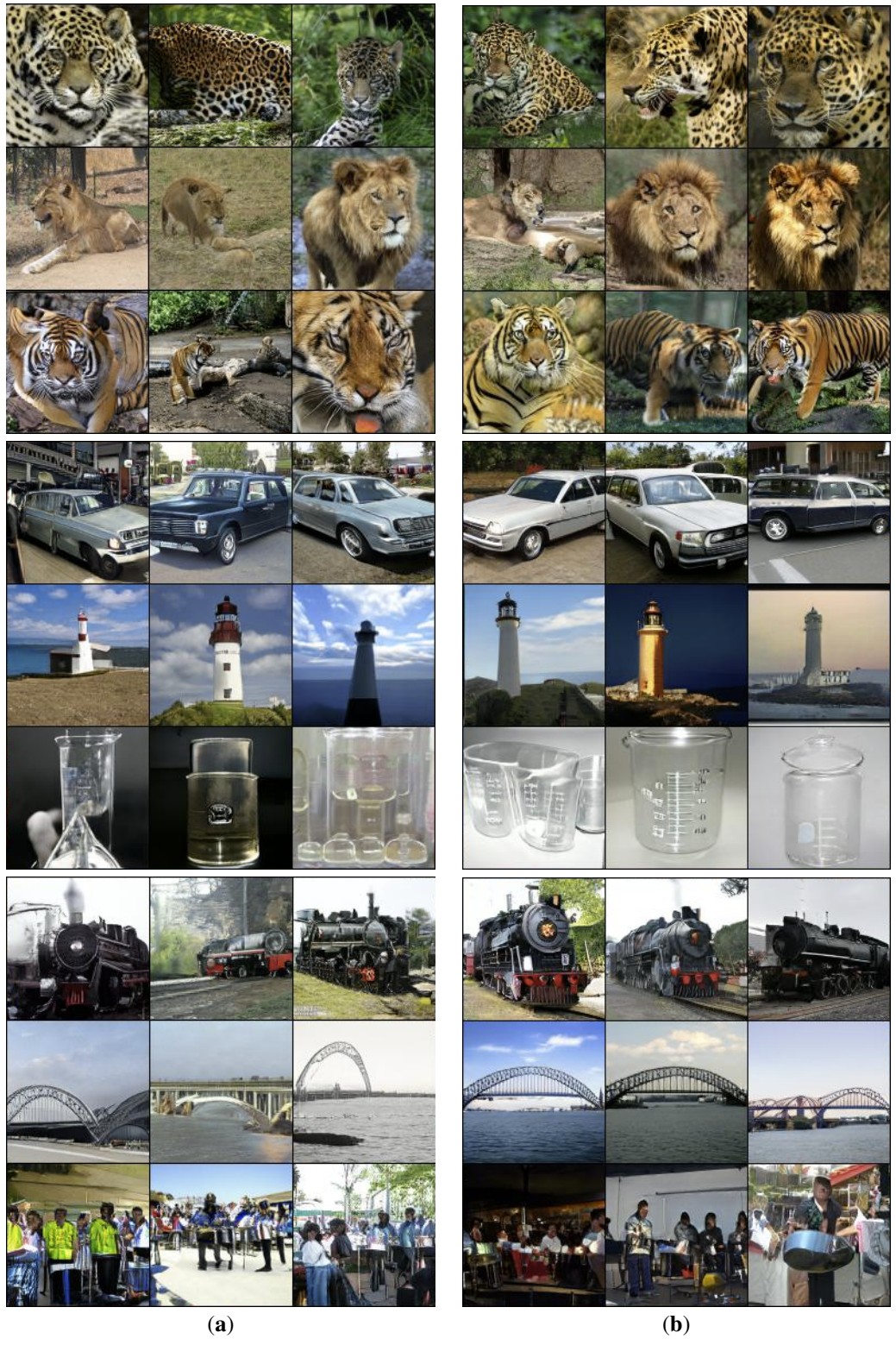

(**a**)                                                                    (**b**)

Figure 7: Samples from BigGAN-deep (**a**) and LOGAN (**b**) with the similarly low FID. Samples from the two panels were draw from truncations correspond to points A, B in figure 3 **b** respectively. (FID/IS: (**a**) 5.04/126.8, (**b**) 5.09/217.0)

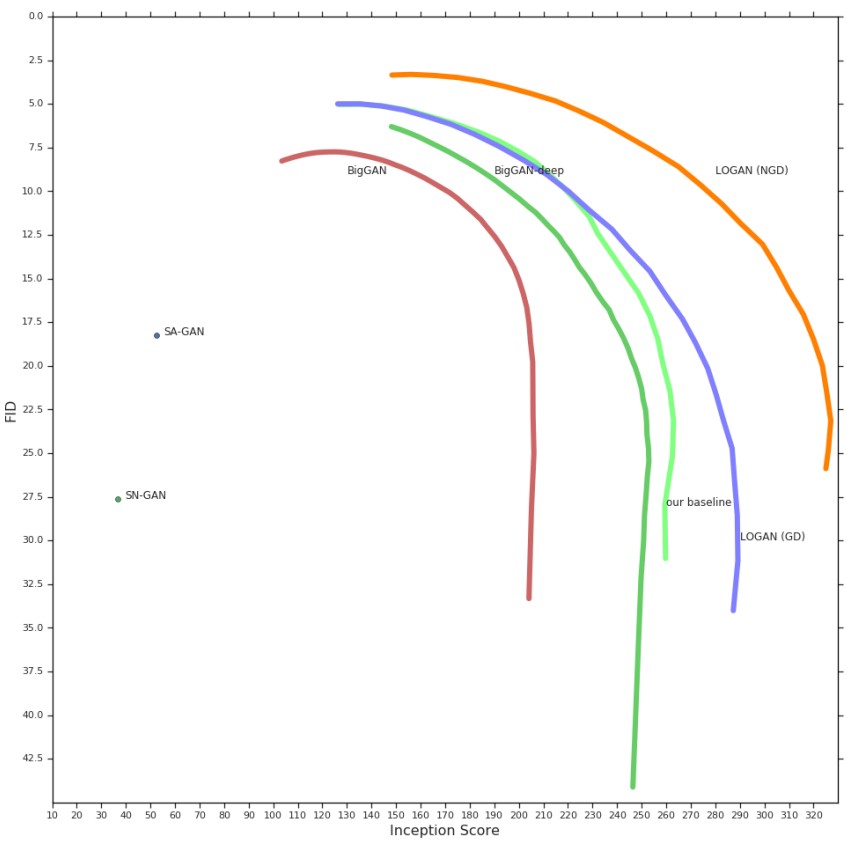

Figure 8: Truncation curves with additional baselines. In addition to the truncation curves reported in Figure 3 **b**, here we also include the Spectral-Normalised GAN (Miyato et al., 2018), Self-Attention GAN (Zhang et al., 2019), original BigGAN and BigGAN-deep as presented in Brock et al. (2018).

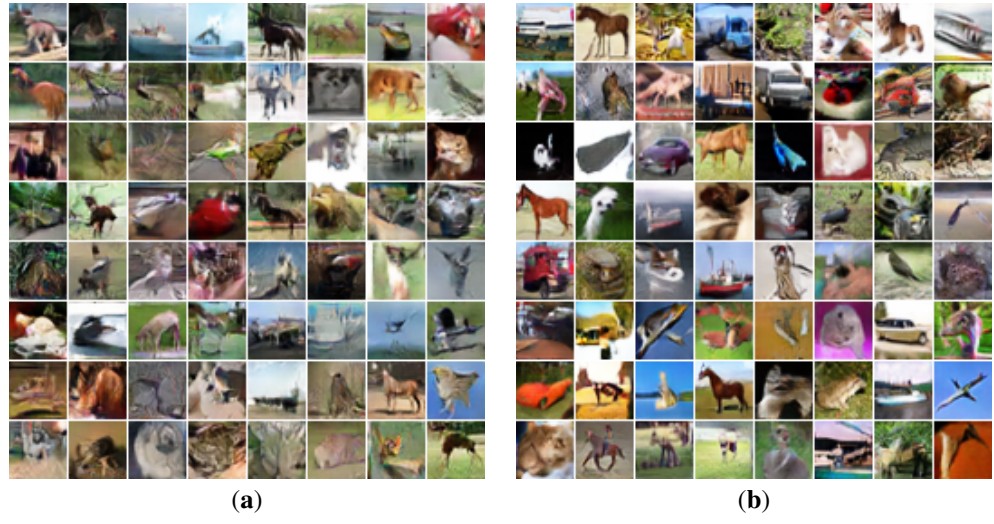

Figure 9: (**a**) Samples from SN-GAN. (**b**) Samples from LOGAN.

