# OpenReview forum: "LOGAN:  Latent Optimisation for Generative Adversarial Networks"
_ICLR.cc/2020/Conference — Reject_

### Official Review · AnonReviewer1 · 2019-10-22
**Official Blind Review #1**

**Rating:** 6

**Review:**

The paper proposes a novel training scheme for GANs, which leads to improved scores w.r.t. state-of-the-art. The idea is to update the sampled latent code in a direction improving the inner maximization in the min-max problem. The work considers both, the gradient direction and a direction motivated by the natural gradient which is shown to yield excellent performance. The overall scheme is motivated as an approximation to the (prohibitively expensive) symplectic gradient adjustment method.

While the work contains some typos, it was easy to read and overall very well written. The experimental results are impressive and clearly validate the usefulness of the approach. Therefore, I recommend acceptance of the paper.

On the theoretical side, I feel some parts can be improved. I'm willing to further increase my rating if some of the following points are addressed:

1. The connection to SGA is a bit hand-wavy, as terms are dropped or approximated with the only reasoning that they are difficult to compute. In some approximations (e.g. Gauss-Newton approximation of the Hessian) one can argue quite well that certain second-order terms can be dropped under some assumptions (e.g. mild nonlinearity, or vanishing near the optimum).

2. I had some troubles Sec 4 related to the natural gradient.
(a) What is p(t | z)? Is it the same p as in (37), Appendix C? I find the argument that the hinge loss pushing D(G(z)) < 0 during training hand-wavy, as for natural gradient p(t | z) should always be a valid distribution. There are also some typos which made it hard to follow the arguments there. It probably should read "an ideal generator can perfectly fool the discriminator" (not perfectly fool the generator).

(b) Maybe it would be easier to directly argue that one wishes to approximate SGA as well as possible, rather than taking a detour through the natural gradient.

3. Why is the increment \Delta z clipped in Algorithm 1? Is there a theoretical justification? If the goal of the clipping is to stay inside the support of the uniform distribution, shouldn't it rather be z' = [z + \Delta z]? A soft clipping (e.g. performing a mirror descent step) might give better gradients.

Typos, minor comments (no influence on my rating):
- In Eq. (5), it should be \partial^2 in the last "Hessian-block" multiplication.
- presribed -> prescribed
- Eqs. (7) and (8) might be easier to parse if one uses a different notation to distinguish between total derivative and partial derivative, i.e., write on the left side df(z') / d\theta. Also, I think it is clearer to write in the last terms in (7) and (8) \partial f(z') / \partial \delta z instead of \partial f(z') / \partial z'.
- In Appendix B.1, shouldn't it be \gamma=(1+\eta)/2 instead of  \gamma=\eta (1+\eta)/2?
- I've found ELU activations to work well in GAN models which involve the Jacobian w.r.t. z. Maybe it can stabilize things here as well.


**Experience Assessment:**

I have published one or two papers in this area.

**Review Assessment: Checking Correctness Of Derivations And Theory:**

I carefully checked the derivations and theory.

**Review Assessment: Checking Correctness Of Experiments:**

I assessed the sensibility of the experiments.

**Review Assessment: Thoroughness In Paper Reading:**

I read the paper thoroughly.

---

> ### Author Response · Authors · 2019-11-08
> **Revision and Clarification**
>
> Thank you for your comments.
>
> We have included your suggested changes in the current draft. Here we directly answer some main concerns:
>
> 1. Symplectic Gradient Adjustment (SGA)
>
> We have rewritten the section on how LOGAN relates to SGA to make it more clear. The main idea is that the latent step in LOGAN introduces a 3rd player into the game who is on the same side as the generator.
>
> Note that the “Hessian” of a game differs significantly from the Hessian of a function: it involves second-order derivatives of multiple functions, corresponding to the losses of the players. The game Hessian thus decomposes into blocks labeled by the “type” of interaction: G+G, G+D, G+Z, and so on. SGA makes use of the anti-symmetric component A of the game Hessian, where there are three kinds of blocks corresponding to second-order interactions between G+D, G+z and D+z respectively.
>
> Second-order interactions between G+D are extremely expensive to compute. Thus, LOGAN only adjusts the gradients based on interactions between G+z and D+z. The partial approximation to SGA is well-founded game-theoretically in that it adjusts the dynamics between 2 of 3 possible pairings of the players — which suffice for a substantial performance gain.
>
> It is currently unclear how approximations like the Gauss-Newton generalize to game Hessians.
>
>
> 2. Natural Gradients (NG) and SGA:
>
> Our main motivation is indeed approximating SGA, and NG is the best way we found to implement the gradient updates in SGA. In other words, SGA does not specify the size of gradient steps, and NG provides an efficient way to adapt the step size under our approximation. We realised that the structure of our submission obscured this storyline, and re-organised it in our current draft.
>
> 3. About the likelihood interpretation of the hinge loss (Appendix C): This section is relatively independent from the rest of the paper. NG has long been used as an approximate optimisation method without considering the probabilistic meaning of the loss. Similarly, SGA does not require the loss to be a likelihood function. We nevertheless provided this section for readers interested in the probabilistic view and acknowledged the limit of this interpretation in the text.
>
> 4. Your comment about the typo is correct, thanks for pointing it out. It should read: “an ideal generator can perfectly fool the discriminator”. This assumption is actually unnecessary, and we removed it. Yes, the likelihood function p(t|z) has the same form as the p in eq. 37. We rewrote this part to make the correspondence more clear.
>
> 5. Thank you for pointing out this typo. The clipping should be after updating z. We have not experimented with many other algorithms for optimising z, and would like to investigate other methods including mirror descent in the future. We will also experiment with ELU activation.

---

### Official Review · AnonReviewer3 · 2019-10-23
**Official Blind Review #3**

**Rating:** 6

**Review:**

Summary:

LOGAN optimizes the sampled latent generative vector z in conjunction with the generator and discriminator. By exploiting second order updates, z is optimized to allow for better training and performance of the generator and discriminator.

Pros:
+ A relatively efficient way of exploiting second order dynamics in GAN training via latent space optimization.
+ A good set of experiments demonstrating the superior performance of the proposed method on both large and small scale models and datasets.

Cons:
- Lack of code

Comments:
All in all, this appears to be a solid contribution to the GAN literature, addressing some of the limitations of CS-GAN [1]. The lack of open source code accompanying the paper (in this day and age) does it a serious disservice. I have already tried and failed to replicate the cifar10 results. There appears to be some nuance in implementation that would probably clear up if the authors release their code along with the paper.

[1] - http://proceedings.mlr.press/v97/wu19d.html

**Experience Assessment:**

I have published one or two papers in this area.

**Review Assessment: Checking Correctness Of Derivations And Theory:**

I assessed the sensibility of the derivations and theory.

**Review Assessment: Checking Correctness Of Experiments:**

I carefully checked the experiments.

**Review Assessment: Thoroughness In Paper Reading:**

I read the paper thoroughly.

---

> ### Author Response · Authors · 2019-11-08
> **Open Source**
>
> Thank you for your comments.
>
> We will open source the CIFAR10 model and training code for the camera-ready version. In the meantime, we would be glad to answer your questions about the implementation, as well as to add any modelling details that might be missing. We appreciate your effort in attempting to replicate the results!

---

### Public Comment · ~Michal_Sustr1 · 2020-01-09
**Question about derivation of SGA dynamics**

The dynamics g from eq. (3) seem wrong. The loss of the discriminator also depend on the training data, but this term is dropped without justification. All of the subsequent derivation then cannot be correct.

In appendix B.1 I don't understand how you can justify the computation of latent step $\Delta z$ as being the third player. The first and last player in eq. (13) come from the same loss function, so their weights are identical, as in the latent optimization. SGA update will however shift those weights differently, so the weights for the first and third player will not be identical anymore.

Thank you for clarification!

---

> ### Author Response · Authors · 2020-01-10
> **reply to questions**
>
> Dear Michal,
>
> Thank you for your questions.
>
> Actually the data-dependent term has already been dropped from eq. 2. This is explained in the text above eq. 2: "We focus on the second term in eq. 1 which is at the heart of the min-max game." Note that the data-dependent term does not participate in SGA (thus does not affect the following analysis), since its derivative with respect to generator parameters is 0.
>
> Seeing \Delta z as the third player provides an intuitive interpretation of the algorithm. Although the first and last players share the same loss, they are not identical: the parameters of the third player are *only* \Delta z. So the derivation following eq.13 simply treats \Delta z as equal of \theta_D and \theta_G.

---

### Decision · Program_Chairs · 2019-12-19

**Decision:**

Reject

**Comment:**

The authors propose to overcome challenges in GAN training through latent optimization, i.e. updating the latent code, motivated by natural gradients. The authors show improvement over previous methods.  The work is well-motivated, but in my opinion, further experiments and comparisons need to be made before the work can be ready for publication.

The authors write that "Unfortunately, SGA is expensive to scale because computing the second-order derivatives with respect to all parameters is expensive" and further "Crucially, latent optimization approximates SGA using only second-order derivatives with respect to the latent z and parameters of the discriminator and generator separately. The second-order terms involving parameters of both the discriminator and the generator – which are extremely expensive to compute – are not used. For latent z’s with dimensions typically used in GANs (e.g., 128–256, orders of magnitude less than the number of parameters), these can be computed efficiently. In short, latent optimization efficiently couples the gradients of the discriminator and generator, as prescribed by SGA, but using the much lower-dimensional latent source z which makes the adjustment scalable."

However, this is not true. Computing the Hessian vector product is not that expensive. In fact, it can be computed at a cost comparable to gradient evaluations using automatic differentiation (Pearlmutter (1994)). In frameworks such as PyTorch, this can be done efficiently using double backpropagation, so only twice the cost.  Based on the above, one of the main claims of improvement over existing methods, which is furthermore not investigated experimentally, is false.

It is unacceptable that the authors do not compare with SGA: both in terms of quality and computational cost since that is the premise of the paper. The authors also miss recent works that successfully ran methods with Hessian-vector products: https://arxiv.org/abs/1905.12103 https://arxiv.org/abs/1910.05852

---

> ### Author Response · Authors · 2019-12-23
> **Response to Meta-review**
>
> The meta-review ignored the positive (two weak accepts) reviews, while raising new criticism that is tangential to our claims. The AC requires additional experimental comparison to the SGA algorithm.  However, we only use SGA to provide a theoretical insight into our method, and don’t claim that our method is an improvement. Our main contribution is empirical, and consists in improving the state of the art by a large margin (>30%) -- this was recognised by the reviewers, but ignored by the AC. As we point out, and reviewer #1 agrees, our method can be viewed as an approximation of SGA which only requires minimal modification of standard GAN training. We mentioned the less straightforward Hessian-vector implementation suggested by the AC in Appendix B.1, and can discuss this in more detail. Further comparison might be an interesting subject for future work, but it’s not a subject of this paper. This point was not raised by either of the two reviewers.
>
> The AC also mentioned two papers which we missed:
>
> [1] Schäfer & Anandkumar, 2019, Competitive Gradient Descent;
> [2] Schäfer, Zheng & Anandkumar, 2019, Implicit competitive regularization in GANs.
>
> The first was just published at NeurIPS 2019 (uploaded to arXiv in May), while the second is a *concurrent* submission to ICLR 2020, and was uploaded to arXiv in October *after* the ICLR deadline. We are happy to cite these papers, but we were not previously aware of them, and could not possibly be aware of the second one as it was not public at the moment of submission. We also note that neither of the original two reviewers mentioned them in their reviews.